# A Review of *Encephalitozoon cuniculi* in Domestic Rabbits (*Oryctolagus cuniculus*)—Biology, Clinical Signs, Diagnostic Techniques, Treatment, and Prevention

**DOI:** 10.3390/pathogens11121486

**Published:** 2022-12-07

**Authors:** Anca-Alexandra Doboși, Lucia-Victoria Bel, Anamaria Ioana Paștiu, Dana Liana Pusta

**Affiliations:** 1Department of Genetics and Hereditary Diseases, Faculty of Veterinary Medicine, University of Agricultural Sciences and Veterinary Medicine Cluj-Napoca, Calea Mănăştur 3-5, 400372 Cluj-Napoca, Romania; 2New Companion Animals Veterinary Clinic, Faculty of Veterinary Medicine, University of Agricultural Sciences and Veterinary Medicine Cluj-Napoca, Calea Mănăştur 3-5, 400372 Cluj-Napoca, Romania

**Keywords:** *Encephalitozoon cuniculi*, encephalitozoonosis, *Oryctolagus cuniculus*, rabbit, diagnostic tests

## Abstract

*Encephalitozoon cuniculi* is a eukaryote, unicellular, spore-forming, obligate intracellular microorganism of the phylum Microsporidia, with domestic rabbits as its main host. Another important species in which this pathogen has been identified are humans, the infection being therefore called a ”zoonosis”. The transmission takes place via the horizontal route or the vertical route, and cell-mediated immunity plays the biggest role in the infected hosts’ protection. Encephalitozoonosis can manifest itself as an acute infection, with neurological signs, renal signs, and ocular lesions, or as a chronic or subclinical infection, which is usually the case for asymptomatic carriers. The diagnostic techniques usually carried out are histological examination, serological tests, and molecular genetic techniques. The treatment of encephalitozoonosis is usually symptomatic, with unrewarding results, and prevention methods include periodical serological screening, prophylactic administration of fenbendazole, and maintenance of a clean environment. The purpose of this article is to review the current data regarding the pathogenesis, host immunity, clinical signs, diagnostic methods, treatment, and prevention methods of encephalitozoonosis in the domestic rabbit, as well as to analyze the prevalence of this disease in different countries of the world.

## 1. Introduction

The phylum Microsporidia consists of eukaryotic, unicellular, obligate intracellular spore-forming microorganisms that were identified over 150 years ago [1]. They are characterized mostly as parasites—more precisely, protozoans—but have been shown to be closely related to fungi. Microsporidians are ubiquitous [2], with infections with this pathogen having economic importance in the silkworm, honeybee, shrimp, and fish industries. The most important species that can be affected is the human [1]; frequently, AIDS patients, the immunosuppressed, and organ transplant patients manifest the disease through a large variety of symptoms. Some of the genera also identified in human infections include *Nosema*, *Vittaforma*, *Brachiola*, *Pleistophora*, *Enterocytozoon*, and *Encephalitozoon* [2]. *Encephalitozoon cuniculi* is part of the aforementioned group of pathogens, with domestic rabbits (*Oryctolagus cuniculus*) as its main host, but also identified in species such as monkeys, foxes, birds, dogs, cats, mice, and humans. The four genotypes of *E. cuniculi* are as follows: genotype I—the rabbit strain; genotype II—the murine strain; genotype III—the canine strain; and genotype IV—the human strain. Even though the names are correlated with the genotypes’ host preference, there seems to be no strict specificity [3]. Pet rabbits infected with encephalitozoonosis can show signs of neurological and renal disease, but the infection can also be subclinical, with a significant number of asymptomatic carriers existing, which is also why diagnosing and preventing the transmission of this pathogen are of the utmost importance [4]. Among all parasitic diseases, encephalitozoonosis has proved to be the one most present in rabbits, according to a recent retrospective study on necropsies carried out in Spain [5]. The high morbidity and the potential zoonotic risk of this disease can directly impact the rabbit industry through economic losses, especially in countries where rabbits have become an important meat source, such as Italy or Egypt [6,7]. The high prevalence of the parasite in rabbit farms in general is most likely due to poor husbandry practices and inadequate prophylactic health measures, while the presence of *E. cuniculi* in younger animals—between 0 and 4 months old—is probably the result of immature immune systems unable to fight the microorganism [5]. Some recent sources [5,7] suggest that young rabbits are more frequently affected than older individuals, and females more than males, but these findings should not be taken as a one-size-fits-all approach. This study aims to review the current data regarding *E. cuniculi* infection in rabbits, from the pathogenesis, host immunity, clinical signs, and diagnostic methods to methods of treatment and prevention of the disease, underlining the significance of limiting the spread of this pathogen among all species.

## 2. Physiopathology

Microsporidia are transmitted through spores, which are generally small oval- or pyriform-shaped structures, with their average size varying between 1 and 4 µm length in mammals. The mature spore consists of an outer layer called the exospore and an inner thicker layer called the endospore, followed by a plasma membrane that surrounds the spore contents. The components of the spore by which microsporidians are identified are the polar filament or tubule, the anchoring disk, the lamellar polaroplast, and the tubular polaroplast. In the center of the spore is the nucleus or diplokaryon, surrounded by the coiled polar filament, localized in a cytoplasm tightly packed with ribosomes. In addition to these components, there is also a posterior vacuole, usually containing a flocculent material [1].

The life cycle of microsporidians that infect humans comprises the infective, proliferative, and sporogonic phases. The infective phase begins at the moment when the spore is fully developed in the host cell [2]. Osmotic pressure and pH changes lead to the extrusion of the polar filament and sporoplasm from the spore, followed by the penetration and injection of the sporoplasm into a new host cell [4], with this process being called the invasion synapse [1]. The proliferative phase then begins, with the injected sporoplasm growing and developing into a meront and dividing itself via merogeny into sporonts inside the host cell’s cytoplasm [2]. The last phase—the sporogonic phase—takes place inside the parasitophorous vacuole inside the host cell’s cytoplasm, where the sporonts differentiate into sporoblasts, which will develop their polar tube and all of the other components and will ultimately turn into mature spores. The rupture of the parasitophorous vacuole and the host cell will determine the dissemination of infective spores into the entire organism of the host [4] via infected macrophages or by their release into the blood [8]. Proteins of both the spores’ walls and the hosts’ cells participate in the infective process [1].

The transmission of *E. cuniculi* in rabbits takes place through two main routes: horizontal infection, and vertical or transplacental infection. The horizontal route usually occurs through oral ingestion of contaminated food or water, and more rarely through inhalation of spores [4]. After the ingestion of the parasite, it goes into the intestinal epithelium, where it replicates, and then the infected macrophages reach the liver, kidneys, central nervous system (CNS), lungs, and heart via circulation. Lastly, the cells in the targeted organs rupture, causing the release of infective spores that will lead to inflammatory and granulomatous lesions [9]. Spores in the urine are intermittently shed after approximately 35 days post-infection and up until 3 months or more [8]. The vertical or the transplacental route, from doe to kit, has also been proven to be another pathway of transmission of *E. cuniculi* in rabbits [4]. Spores of *E. cuniculi* were identified in the ocular structures of the offspring—more precisely, in the eye lens—through molecular genetic techniques. This predilection site causes lesions of cataracts or uveitis. It is believed that the moment of infection is during the first trimester of gestation in the rabbit’s embryological development, when the lens placode and the lens capsule form, but spores can also remain in the anterior lens capsule [10]. Infection through rupture of the lens capsule due to a disruption of normal epithelial function is also another explanation of this transmission route, where a sudden release of parasitic proteins that initiate a cellular immune response against normal lens proteins occurs, leading to the clinical lesion of unilateral phacoclastic uveitis [11]. Other experimental transmission routes consist of traumatic transmucosal, intravenous, intrathecal, and rectal infection [4].

## 3. Host Immunity

When it comes to the immunity of the host infected with *E. cuniculi*, there are two major types that have previously been identified, mainly in murine models: cell-mediated immunity, and humoral immunity.

The cell-mediated response of the host has proven to be the superior one; studies on mice showed that this immunity not only improved the hosts’ resistance but also reduced lethal *E. cuniculi* infection [4]. Both CD4^+^ and CD8^+^ T lymphocytes play an important protective role in the oral ingestion of the pathogen [8]. Jeklova et al. [8] conducted a study on experimentally infected rabbits, where antigen-specific lymphocyte proliferation in the spleen was phenotyped using a carboxyfluorescein succinimidyl ester (CSFE) stain. After 2 weeks post-infection, the proliferation of CD4^+^ T lymphocytes was higher than that of CD8^+^ T lymphocytes; at 4 weeks post-infection, both lymphocyte types had a similar amount of proliferation; while at 6 and 8 weeks post-infection, the proliferation of CD8^+^ T lymphocytes exceeded that of CD4^+^ T lymphocytes [8]. Other essential structures implicated in the cell-mediated immunity of this disease are cytokines, secreted by T cells as an immune response, which direct macrophages to phagocytose infected cells [4]. IFN-γ is one of these important cytokines that acts as an activator of macrophages, leading them to produce toxic oxygen metabolites capable of destroying the phagocytosed *E. cuniculi* spores. IFN-γ also seems to offer immunological anti-microsporidial protection regardless of the route of infection [8]. In the study by Jeklova et al. [8], molecular genetic techniques were used to identify high amounts of IFN-γ mRNA in the spleen, mesenteric lymph nodes, and Peyer’s patches post-infection, while the amounts of IL-4, IL-10, and IL-17 mRNA presented fluctuations during the 8 weeks post-infection. An exception was the small intestine, where IL-4, IL-10, and IL-17 mRNA exceeded the amount of IFN-γ mRNA in weeks 4 and 6 post-infection. Therefore, the Th1 immune response (IFN-γ-secreting T cells) was dominant in the spleen, mesenteric lymph nodes, and Peyer’s patches in comparison to the Th2 immune response (IL-4- and IL-10-secreting T cells) and to the Th17 immune response (IL-17-secreting T cells), while Th17 exceeded the Th1 response in the small intestine [8]. Intraepithelial lymphocytes (IELs) are another cell type implicated in the cell-mediated immunity, by producing high amounts of IFN-γ and displaying intensive cytolytic processes that can prevent parasite multiplication [8]. Natural killer (NK) cells also play a role by secreting IFN-γ to activate the phagocytic function of macrophages, while also mediating innate responses through perforin-mediated lysis of infected cells [4]. Studies show that dendritic cells may also have an essential protective mechanism, since dendritic-cell-deficient mice were susceptible to reinfection with *E. cuniculi*, and sera transfer from young mice exposed to the parasite offered protective immunity [4].

Humoral immunity is characterized by a production of antibodies—more specifically, immunoglobulin M (IgM) and immunoglobulin G (IgG)—by B cells or B lymphocytes, as a response to the host organism being infected. In the case of *E. cuniculi* infection in rabbits, it appears that this type of immunity does not offer sufficient protection against reinfection, despite the fact that the antibodies can persist for the hosts’ entire life after exposure to the pathogen. Humoral immunity cannot provide protection all by itself, as demonstrated in a study on athymic mice exposed to *E. cuniculi* infection, which did not survive the disease despite administration of immune serum [4]. In the study of Jeklova et al. [8], both IgM and IgG had significantly higher titers in the infected rabbits compared to the non-infected rabbits. Other similar studies on experimentally infected rabbits analyzed the IgM and IgG titers throughout a 68-day period post-infection. The IgM titers, which indicate an acute infection, were identified in high amounts after the first 20–30 days post-exposure, followed by a decrease over the next 8–10 days. On the other hand, the IgG levels were either at a plateau throughout the entire time, in a linear elevation, or were not at all detectable at the end of the 68 days, leading to the conclusion that the different responses were influenced by either the individual variation in immune response or the *E. cuniculi* exposure load. It is safe to say that measuring the IgM or IgG titers of an individual does not provide exact information about the moment of exposure to *E. cuniculi*, as well as that cell-mediated immunity remains the superior mechanism in terms of protection against the disease [4].

## 4. Clinical Signs

Encephalitozoonosis in rabbits can manifest itself in different ways depending on the immunity of the host, with immunocompetent individuals having a mild or subclinical form of the disease, while immunocompromised patients show severe clinical signs with possible fatality [12]. The disease can be acute or chronic, the latter case being clinically harder to observe. The clinical signs shown in *E. cuniculi* infection in rabbits have a strong connection to the main affected organs: the central nervous system (CNS), the kidneys, and the eyes [13].

Vestibular disease is the most common manifestation seen in acute cases of encephalitozoonosis in rabbits as a result of CNS lesions [13]. Clinical signs can vary in severity, from torticollis or head tilt, ataxia, nystagmus, hemiparesis or paresis, tremors, and seizures to longitudinal rolling and hindlimb paralysis with urinary incontinence [12,13,14]. Renal insufficiency is the result of a chronic infection; clinical signs include polyuria, polydipsia, pollakisuria, azotemia, weight loss, and cystitis, but these are usually hard to observe [12,14]. Damage to the eye globe can result from the parasite invading the eye lens, inducing inflammation and the spontaneous rupture of the anterior lens capsule at its thinnest point, which causes a release of lens material into the anterior chamber of the eye globe, leading to phacoclastic uveitis. This kind of lesion is usually unilateral [13]. In addition to uveitis, secondary glaucoma and cataracts can also occur [12]. Taking the variety of clinical signs and their low specificity into account, most authors are of the opinion that ante-mortem diagnosis of the disease remains a real challenge [4].

## 5. Diagnostic Methods

Finding the most efficient method of diagnosing *E. cuniculi* in rabbits has been a true challenge over the last several decades. The main methods are histopathologic diagnosis, serological diagnosis, and diagnosis through molecular genetic techniques [4]. Other paraclinical tests can be helpful in establishing a prognosis, such as blood biochemistry to evaluate the renal parameters or computerized tomography (CT) scan to identify the extent of cerebral lesions or differentiate them from otitis [15].

### 5.1. Histopathological Diagnosis

The lesions and the presence of parasitic spores found in the postmortem examination are probably the most reliable markers in establishing the diagnosis of *E. cuniculi* in rabbits. The findings of this exam are usually in the main targeted organs—the brain, kidneys, and eyes—but other tissues such as the liver, lungs, heart, and spleen have also been found to be affected [4].

Upon macroscopic examination, congestion of the meningeal and cerebral vessels can be noticed, together with an acute severe multifocal necrosis in the cerebrum [7]. The kidneys are frequently pale and enlarged, with fibrosis indicated by capsule adherence to the parenchyma [4,7]. The eye globes are usually unilaterally affected, with focal uveitis in the anterior chamber, the presence of lens opacity, and possible increased thickness of the cornea [16].

Histological exams applying hematoxylin and eosin (HE) and Giemsa stains generally reveal lesions of granulomatous meningoencephalitis and chronic interstitial nephritis; however, these are not pathognomonic [4]. The brain tissue presents perivascular cuffs in all cerebral lobes, composed of plasma cells, lymphocytes, and macrophages, also called granulomatous lesions [17]. Multifocal gliosis and glial nodules with neuronophagia and neuronal degeneration have also been observed [7]. Mature *E. cuniculi* spores or cyst-like aggregates can be identified inside parasitophorous vacuoles within the cytoplasm of perivascular macrophages and in neurons, while extracellular spores can be seen all throughout the cerebral cortex and hippocampus [7,18].

Located in the renal tissue, lesions such as mesangial proliferative glomerulonephritis with vacuolization and necrosis of the epithelium lining the convoluted tubules and collecting ducts can be identified [7,18]. In severe cases, the presence of fibrosis and interstitial deposition of collagen—associated with a mixed inflammatory infiltration composed of plasma cells, lymphocytes, and macrophages—is also noted. Atrophy and thickening of the basement membrane are some of the glomerular lesions [18]. The presence of spores has been identified in the cytoplasm of tubular epithelial cells, as well as extracellularly in the renal tubules [7,18].

The affected eye globe generally presents lesions of the lens, with the anterior lens capsule ruptured or thinned in some areas and destruction of the lens fibers, followed by degeneration and cellular necrosis [7,16]. In addition to these changes in the lens, other lesions that can occur include corneal edema, epithelial ulceration, and endothelial necrosis, together with mononuclear cells mixed with fibrin infiltration in the corneoscleral trabecular meshwork, in the junction between the cornea and conjunctiva, in the iridocorneal drainage angle, and in the posterior chamber of the eye [7]. Other ocular changes seen previously include loss of epithelial integrity of Bowman’s capsule and Descement’s membrane, edema of the iris with degeneration of the posterior epithelium, and atrophy and detachment of the retina. Parasites have been found attached to the anterior lens capsule or in the ganglion cells of the retina [7,16].

Other tissues with modifications due to encephalitozoonosis in rabbits include the liver, with mild degeneration of hepatocytes in the centrilobular area and periportal mononuclear cellular infiltration, and the lungs, where mononuclear cellular infiltrates and cell degeneration are less abundant, but hyperemic capillaries stand out [7,8]. In the spleen, lymphocyte infiltration together with hyperemia of the red pulp can be observed [8].

With regard to the techniques generally used for histopathological diagnosis of *E. cuniculi* in rabbits, a lot of different stains have been tested. Gram staining has proven to be effective, with the identification of Gram-positive microsporidia in the parasitophorous vacuoles of cells in the targeted organs. In order to differentiate the microorganisms found from Gram-positive bacteria, the type of inflammation and morphology need to be taken into account, but other stains can also be used in cases of uncertainty [4]. For the identification of mature spores, periodic acid–Schiff (PAS) staining, Ziehl–Neelsen (ZN) staining and acid–fast trichrome (AFT) staining have also shown good results [4,16]. Spores within the cytoplasm of macrophages, mesangial cells, and epithelial cells could be identified using ZN staining [19]. Moreover, ZN staining showed increased sensitivity in detecting spores in inflammatory lesions compared to HE staining [4], and also compared to AFT staining [19]. While HE staining only revealed inflammatory lesions, ZN staining managed to detect histological positivity even in those tissues with a small load of parasites [17].

Immunohistochemistry is another technique that can be used in the diagnosis of encephalitozoonosis in rabbits, where spores are detected based on the reaction with specific anti-*E. cuniculi* monoclonal antibodies (EC11C5) [11]. The tissues found to have the highest load of spores were the cerebrum, medulla oblongata, and leptomeninges, but the parasite was also detected in the liver, kidneys, lungs, heart, mesenteric lymph nodes, spleen, and intestines [8]. *E. cuniculi* was present as round-to-ovoid organisms of granular appearance and localized predominantly in the extracellular space, but some of them also intracellularly. The most affected eye structures were the periocular connective tissue, the sclera, and the cornea, and to a lesser extent the iris, the retina, and the lens [11].

Transmission electron microscopy (TEM) is another option, where all of the proliferative stages (i.e., meront, sporont, sporoblast) of *E. cuniculi* can be seen in the targeted organs. In Morsy et al.’s study [7], all of the different stages could be detected intracellularly in brain tissue, while the renal tissue only revealed the sporont stage and mature spores.

Even if severe histological lesions can be identified in the brain and kidneys, it has been proven that they are not necessarily associated with the manifestation of clinical signs. Therefore, the severity of the histological lesions cannot determine *E. cuniculi* as the causative agent of the disease [19].

### 5.2. Serological Diagnosis

Serological testing in the case of a suspected *E. cuniculi* infection is one of the most widely utilized ante-mortem diagnostic methods, as the options are quite limited. Through this method, the specific antibodies IgM and IgG are titrated; however, a positive result only indicates whether there was a previous exposure of the animal to *E. cuniculi*, with no exact information as to when this occurred. It is recommended that more than one titer be obtained from an individual in order to establish the exposure to the microorganism. Clinical signs are not necessarily correlated with a high antibody titer, and only a negative result can rule out encephalitozoonosis [4]. The techniques generally used are enzyme-linked immunosorbent assay (ELISA), carbon immunoassay (CIA), indirect fluorescent antibody test (IFAT), Western blot analysis, and C-reactive protein (CRP) measurement [18].

Studies conducted on the serological diagnosis of *E. cuniculi* in rabbits reported a significant correlation between the qualitative results of ELISA and IFAT methods, as well as between ELISA and CIA and between IFAT and CIA [20,21]. Quantitative titers can also be determined using ELISA [21]. The CIA technique is a fast, easy, and low-cost test that can be used for screening rabbit populations for encephalitozoonosis; however, only IgG can be determined through this method [20]. Western blot analysis shows a slightly higher sensitivity in the detection of IgM and IgG compared to ELISA, but due to its lack of standardization and labor-intensive features it seems to be an unsuitable routine test. Measurement of CRP only serves as an indicator of systemic inflammation in conjunction with seropositivity, if existing, but increased values are not specific to encephalitozoonosis [22].

Determining the IgM and IgG antibody titers in a suspected case of encephalitozoonosis is an indicator of a humoral response of the host organism, but these do not show anything more than the fact that an exposure to the parasite took place [4,8]. Although there is a differentiation between the IgM and IgG titers regarding their increase and telling whether the infection is active, latent, or old and cured, they cannot ascertain whether *E. cuniculi* is the causative agent of the disease [13,19]. The titers considered to be positive differ from author to author. Most studies report that a high IgM titer indicates an early or acute infection, high titers of IgG indicate a chronic or latent infection, and both IgM and IgG in high amounts simultaneously indicate an active infection—either acute, a reactivated infection, or reinfection [23].

During the course of infection, serum levels of IgM decrease, while those of IgG increase [24]. A study on an experimental oral and ocular infection model noted that high IgM and IgG titers were identified until 18 weeks post-infection, with the highest IgM spike within the first week and the highest IgG spike by the third week. This suggests that rabbits with acute *E. cuniculi* exposure show high IgM titers between 0 and 35 days post-infection that can persist for up to 18 weeks—a time frame in which the animal can eliminate the spores via urine, which makes it very important that they are isolated from the rest of the population. Even so, after 5–18 weeks, the IgM titers will start to decrease and will no longer be helpful in establishing whether the animal is suffering from an acute infection [4].

It is important to highlight that a negative result does not automatically mean that exposure to *E. cuniculi* has not taken place. A recently exposed immunocompetent rabbit can show negative titers for up to 2 weeks post-infection; therefore, only a second negative IgG titer in the clinically healthy animal 3 weeks later can confirm non-exposure to the pathogen. If a clinically healthy individual presents a single positive IgG titer, this can mean either that it is an early infection, a chronic infection, and/or previous infection and recovery from encephalitozoonosis. In the presence of neurological clinical signs and two negative IgG titers 3 weeks apart, *E. cuniculi* can be ruled out as the causative agent of the disease [4].

The significance of age at the time of testing is also essential to take into account. Young rabbits have maternal antibodies passed from the mother up until 4 weeks of age, during which time they will test seropositive, followed by the disappearance of the antibodies from 4 to 8 weeks of age, where they will be seronegative [25]. This could be explained by the fact that some studies [6,25] report a lower prevalence of the disease based on antibody detection in rabbits less than 4 months old compared to those over 4 months old, concluding that any rabbit between the age of 4 and 8 weeks is likely to show a false seronegativity.

Numerous studies have been conducted on the serological diagnosis of *E. cuniculi* in rabbits of multiple origins and with different clinical statuses, whether they are clinically healthy, or with specific or non-specific encephalitozoonosis signs (Table 1).

### 5.3. Diagnosis through Molecular Genetic Techniques

Molecular genetic techniques of diagnosing *E. cuniculi* in rabbits have been used more and more in recent years and still require more research in order to establish their efficacy. The samples that have been tested previously through these methods include urine, feces, cerebrospinal fluid (CSF), and organs collected through necropsies, such as brains, kidneys, eye lenses, livers, lungs, hearts, and spleens [19,31]. Specific DNA extraction is carried out, followed by amplification using either conventional polymerase chain reaction (PCR), nested PCR, or RT-PCR [19].

The first step of this diagnosis is the extraction of DNA from the collected materials, with the organ samples being deposited beforehand at −20 °C. This is usually performed according to the instructions of the kit manufacturer, with small modifications depending on the author. Because of the thick wall of microsporidians, a pretreatment is usually required in order to destroy the wall so that the DNA is released, which is performed with either liquid nitrogen, mechanical disruption with glass beads, and/or digestion with proteinase K. The only exception from this are the eyes with phacoclastic uveitis, where the spore load is sufficiently high that DNA can be extracted and amplified easily [4,19].

PCR amplification follows after obtaining the DNA isolates, the type of PCR reaction being different depending on the sample tested and the author. In the case of conventional PCR, the primers ECUNF (5′-ATGA-GAAGTGATGTGTGTGCG-3′) and ECUNR (5′-TGCCATGCACTCACAGGCATC-3′) have been used to target the 549 bp fragment of the 16S small subunit ribosomal RNA gene [16,19]. The conventional reaction has been successfully used in eye lens material because of the high spore load [19]. When it comes to body excretions and other postmortem organs, nested PCR is preferred, as it has a higher sensitivity, where the primers P2 (5′-TTGCGGGATGAGCAGTAGCTGCG-3′) and M2 (5′-TGCTGCCACAAACA-CAACCCG-3′) are used for the first PCR reaction, followed by a second (nested) PCR reaction using the ECUNF and ECUNR primers [19]. Brain tissue had the highest sensitivity in *E. cuniculi* detection using nested PCR [19]. RT-PCR has been used for the detection of *E. cuniculi* in urine, but no concrete information on its sensitivity has been offered [32]. Diagnosis using PCR reactions in urine, feces, and CSF is problematic and, unfortunately, has a low sensitivity according to most authors. Spores’ excretion is short and intermittent in urine and feces, which makes it hard to rely on in detecting *E. cuniculi* DNA from these materials [4,31]. Diagnosis from CSF is just as disputable, since previous studies proved a low prevalence in this type of bodily fluid, concluding that these could very well give us false negative results. Therefore, what needs to be noted is that spore detection using PCR will be highly dependent on the stage of infection (i.e., acute, latent, or reinfection) and the spore load within the animal [4].

For the genotyping of the *E. cuniculi* isolates, semi-nested PCR can be used, targeting the polar tube protein (PTP) gene with the first pair of primers PTPF (5′-GCAGTTCCAGGCTACTAC-3′) and MPTPR (5′-CATGACAGGGTTTGGAATCT-3′) to amplify a 423 bp fragment for genotypes I and II and a 345 bp fragment for genotype III. For the second PCR, the pair of primers PTPF and PTPR (5′-AGGAACTCCGGATGTTCC-3′) can be utilized to target a 363 bp fragment for genotypes I and II and a 285 bp fragment for genotype III [33].

Different studies report the prevalence of encephalitozoonosis in rabbits using molecular genetic techniques, with the highest being detected in eye lens and brain tissues (Table 2).

### 5.4. Other Paraclinical Tests

There are several tests that can help assess the prognosis of an acute infection or guide us in the direction of a differential diagnosis, but these certainly cannot give us a definitive *E. cuniculi* diagnosis in rabbits. Some options include hematology for evaluating infection markers, serum protein electrophoresis for evaluating the specific proteins and biochemistry, and kidney ultrasound for assessing renal function [4,9]. In addition to these, a CT scan can be performed to help us evaluate the cerebral damage or lead us to a differential diagnosis of otitis interna/externa [15].

Hematology from rabbits infected with *E. cuniculi* that showed signs of renal disease revealed changes such as increased absolute heterophil counts and a low hematocrit (<33%)—changes that can, however, go unnoticed [9]. Biochemistry is a rather more helpful tool, where several studies have reported changes in renal values—especially in acute encephalitozoonosis infections. In one study, the values of blood urea nitrogen (BUN), creatinine, alkaline phosphatase (ALP), cholesterol, phosphorus, and glucose were significantly higher in seropositive rabbits compared to seronegative rabbits [35]. Another study also reported higher values of urea concentration and ALP in seropositive animals [10], but decreased phosphorus and potassium levels in affected rabbits with signs of renal disease have also been reported [9].

Serum protein electrophoresis has been used for the measurement of the albumin-to-globulin ratio in rabbits infected with *E. cuniculi*. A comparison between infected and clinically healthy rabbits revealed that the affected ones usually had a lower albumin-to-globulin ratio, a lower β-globulin fraction, and a higher γ-globulin fraction [9]. Another study on rabbits with suspected encephalitozoonosis also revealed a significant increase in γ-globulin fraction and a low albumin-to-globulin ratio; however, in the rabbits with clinical signs that were not suspected of the disease, the β-globulin fraction also showed an increase [36]. This can only be used as an accessory tool in the diagnostic process [36].

## 6. Differential Diagnosis

Keeping in mind that the clinical signs of encephalitozoonosis in rabbits are not really specific to the disease and that establishing a diagnosis can be quite hard and expensive for the owners, looking into other similar pathologies is essential in making a correct diagnosis. The causes of neurological signs affecting the CNS can be infectious and non-infectious. The infectious pathogens that cause central vestibular syndrome are of bacterial, viral, protozoal, and parasitic origins. The bacterial agents are the most frequently encountered, such as *Pasteurella multocida*, *Staphylococcus aureus*, *Pseudomonas aeruginosa*, *Escherichia coli*, and *Listeria monocytogenes*, while among the less frequently diagnosed pathogens are human herpesvirus 1, *Toxoplasma gondii*, and rabies. Non-infectious causes of CNS signs in rabbits include lymphosarcoma or metastatic neoplasia, lead toxicity, hydrocephalus, cerebral infarcts, hepatic encephalopathy, enterotoxemia, and sepsis. Focal spinal disease or disease anywhere along the cerebral–brainstem–spinal cord axis can also be etiologies of paresis, plegia, or ataxia [4].

In cases of peripheral vestibular syndrome, a few differential diagnoses include bacterial otitis interna/externa, traumatic rupture of the tympanic bulla, aminoglycoside-mediated ototoxicity, and idiopathic vestibular disease [4].

Although renal impairment is rarely manifested clinically in rabbits with the disease, weight loss and emaciation despite a normal food intake can be associated with chronic azotemia. Ultrasonographic changes of the kidneys in acute onset have been observed in rabbits with chronic renal failure [4].

The differential causes of ocular disease are lens-induced uveitis secondary to geriatric cataract formation, and bacterial uveitis—a lesion that is usually bilateral and can be encountered in *Pasteurella multocida* infection [4,20].

## 7. Treatment and Prevention Methods

Therapy can be just as challenging as establishing a diagnosis of encephalitozoonosis in rabbits, since this disease has no specific cure and acute cases in immunocompromised animals usually have a deadly outcome [19]. The most frequently used medications are symptomatic and antiparasitic drugs, with the prognosis being guarded—especially in severe brain and kidney lesions that are irreversible [13]. Treating chronic cases of encephalitozoonosis can be easier, but still with no guarantee of a full recovery [4].

The parasitic spores cause the inflammation of the ruptured host cells that they have invaded, which is why treatment goals consist of reducing spore proliferation and migration, reducing spore-mediated inflammation, and management of severe neurological signs and concurrent disease [4].

Since *E. cuniculi* is part parasite and part fungus, first on the medication list are antiparasitic and antimycotic drugs. The study of Wei et al. [37] regarding anti-microsporidial therapy in humans indicates that benzimidazoles have anti-inflammatory, anti-hypertensive, anti-bacterial, anti-parasitic, and anti-fungal effects, with albendazole having a high success rate. The same study reported the use of fumagillin, from the terpenes class, with a successful outcome in *E. cuniculi* infection in humans [37]. On the other hand, La’Toya et al. [4] outlined that albendazole treatment in humans was followed by relapse of urinary spore shedding. The use of albendazole in the case of infected rabbits has proven to be unfavorable, having embryotoxic and teratogenic effects as well as causing liver disease with prolonged use of the drug [4].

A better option for antiparasitic medication with fewer adverse effects in rabbits is fenbendazole—also from the benzimidazoles class—recommended both for treatment of the disease and as prophylactic administration. The preventive use of fenbendazole prior to experimental infection with *E. cuniculi* in rabbits showed seronegativity after 21 days post-infection and the absence of spores in brain tissue upon postmortem analysis. Most authors use the off-label treatment with fenbendazole orally dosed at 20 mg/kg body weight daily for 28 days, with the success rate in acute or chronic cases of encephalitozoonosis being lower than in prophylactic administration [4,9].

In the treatment of severe systemic inflammation, steroids have shown promising results, although their use is controversial due to the anti-inflammatory versus immunosuppressive effects. The administration of dexamethasone at 2 mg/kg body weight intramuscularly for three doses every 6 h in experimentally infected rabbits caused a significant decrease in inflammatory activity of the organism, but at the same time led to acute immunosuppression that exacerbated and/or caused clinical signs of encephalitozoonosis. In cases of cerebral trauma, the use of dexamethasone is also not recommended, because this can worsen the clinical signs, and a therapeutic effect has not been achieved in chronic infection [4]. Another study recommends the use of dexamethasone for up to three doses at 0.1–0.2 mg/kg body weight subcutaneously, once every two days [9]. Non-steroidal anti-inflammatory drugs are an alternative to steroid therapy, but these should be used with caution as they act on the kidneys—a target organ of *E. cuniculi* [4].

Rabbits with signs of neurological disease (e.g., head tilt, seizures, rolling) should receive small doses of diazepam at 0.5 mg/kg body weight subcutaneously/intramuscularly/intravenously or midazolam at 0.07–0.22 mg/kg body weight intramuscularly/intravenously administered as a mild sedative. Even though rabbits are not capable of vomiting and lack the vomiting center in the medulla oblongata, antiemetic drugs such as metoclopramide (0.5 mg/kg body weight orally/subcutaneously, three times a day), prochlorperazine (0.2–0.5 mg/kg body weight orally, three times a day), or meclizine (12.5–25 mg/kg body weight orally, three times a day) have been used in cases of severe torticollis [9].

In acute manifestations of encephalitozoonosis in rabbits, in addition to fenbendazole administration, broad-spectrum systemic antibiotherapy is recommended for any concurrent or secondary bacterial infection. The most frequently used substances are trimethoprim–sulfamethoxazole (15–30 mg/kg body weight orally, two times a day) or enrofloxacin (10 mg/kg body weight orally, two times a day) for 7–10 days [9].

In the study of Künzel et al. [38] on treatment protocols for *E. cuniculi* in rabbits, it was reported that a combination of fenbendazole, oxytetracycline, enrofloxacin, and dexamethasone or prednisone showed a 54.2% clinical recovery rate. Another study [39], where the clinical efficacy of different combinations of fenbendazole, oxytetracycline, and steroid therapy was evaluated, the results demonstrated no significant differences in reducing neurological signs or between the short-term and long-term survival. The rabbits that received fenbendazole proved to be 1.6 times more likely to survive until day 10 compared to the ones that did not have this medication in their protocol [39].

Ocular lesions of encephalitozoonosis in rabbits—more specifically, uveitis—have been treated conservatively with systemic administration of dexamethasone and oxytetracycline together with topical ophthalmic dexamethasone and tetracycline. In cases of focal granuloma and persisting cataracts, surgical removal of the lens by phacoemulsification is another option to consider. It is important to recall that dexamethasone, with an immunosuppressive effect, can be absorbed systemically through ocular administration; therefore, the use of non-steroidal anti-inflammatory drugs is preferred when the renal biochemical values are in their reference ranges [4]. Other topical ocular substances that have been used include fusidic acid or dexamethasone/neomycin/polymyxin B [13].

The in vitro efficacy of various other drugs—such as fumagillin, sparfloxacin, polyoxin D, and nikkomycin Z—has been demonstrated, but their clinical use in rabbits has not yet been documented [9].

Other aspects to be taken into consideration in the acute disease of rabbits are fluid therapy, assisted feeding if needed, prokinetic drugs, and a stress-free environment with protection against self-inflicted lesions that can occur due to neurological manifestations [4,9].

The prevention of *E. cuniculi* infection in both rabbits and other susceptible species—especially in humans—is of the utmost importance. Because of the wide spread of the organism and all of its characteristics detailed in this study, this is not the simplest thing to accomplish. The first and most essential step would be setting up *E. cuniculi*-free rabbit breeding colonies, which unfortunately for breeders can require a lot of time and expense. This should be achieved through periodic serological testing of rabbit populations, with the isolation of healthy seronegative individuals from the others in separate cages. Especially in breeding facilities, the testing of antibody levels should be performed on young rabbits every 2 weeks for 2 months, with the removal of all seropositive animals. Only rabbits that show negative results for an entire month should be used for future breeding, followed by monthly serological testing to confirm their disease-free status [9]. It is possible that once the parasite enters a colony, using the “all-in and all-out” system will be necessary in order to remove the entire population and perform a thorough disinfection of the environment before introducing new rabbits. Moreover, using rabbit breeds from encephalitozoonosis-free rabbitries—such as lop-eared and rex rabbits, which are thought to be more resistant to the disease—would be ideal to form the new livestock [40].

For rabbits that are not assured to come from *E. cuniculi*-free colonies, the best way of reducing the spread of infection would be prophylactic administration of fenbendazole orally or in the food, using the same 28-day protocol mentioned earlier. This should also be applied for newly acquired animals before putting them in contact with other rabbits [9]. What needs to be kept in mind is that this method will not guarantee that future infections cannot appear [9].

Supplementation of the diet with probiotic bacteria is also thought to help by increasing the immune response and by forming a defense against pathogen infection [41]; however, there are no studies documented on their use for the treatment of encephalitozoonosis.

The resistance of *E. cuniculi* in the environment is high, but the spores may be destroyed using certain disinfection protocols, such as 0.1% bleach with a 10 min contact time or ethanol (70%) with a 30 s contact time. Sodium hydroxide (1%), formaldehyde (0.3%), and hydrogen peroxide (1%) are other options that can kill spores with a contact time of 30 min [4].

Other prophylactic measures that should be applied are reducing of rabbits’ contact with other rabbits or even wildlife, thorough disinfection of the environment using substances such as those mentioned above, appropriate hygiene through hand-washing and disinfection after handling rabbits, and the use of raised food dishes and water bottles that reduce the risk of urine contamination [9,40].

## 8. Conclusions

*E. cuniculi* is a eukaryotic, unicellular, obligate intracellular and ubiquitous parasitic organism from the phylum Microsporidia, with the domestic rabbit (*Oryctolagus cuniculus*) being its main host, but also transmissible in numerous other species—most importantly, in humans. The transmission in rabbits takes place either via the horizontal route, through contaminated water and food or, rarely, through spore inhalation, or via the vertical route, from doe to kit, through the ocular structures. The hosts’ superior protection mechanism is cell-mediated immunity provided by the CD4^+^ and CD8^+^ T lymphocytes, followed by humoral immunity, where specific IgM and IgG antibodies are produced. The clinical signs in infected rabbits are mainly of a neurological, renal, and ocular nature, but the disease can also be subclinical, with a significant number of asymptomatic carriers. The diagnosis of encephalitozoonosis through histopathology reveals lesions of granulomatous meningoencephalitis and chronic interstitial nephritis together with spores in the targeted organs. Serological testing is the most widely used ante-mortem diagnosis tool, where specific IgM and IgG titers are detected, but with no value of certainty. Diagnosis through molecular genetic techniques is becoming more popular in recent years, but with a questionable sensitivity of the method. Conventional PCR, nested PCR, and RT-PCR are the methods through which specific DNA has been identified previously—especially from urine and postmortem organs. The treatment of infection usually includes fenbendazole, systemic antibiotherapy, supportive therapy, and the treatment of eye lesions with anti-inflammatory drugs or even surgical intervention, but with a variable rate of success. Prophylactic administration of fenbendazole in rabbits, periodic serological testing of rabbit populations, and maintaining a clean environment are the best ways to reduce the transmission of this widespread pathogen, not only among rabbit populations but, more importantly, in humans, in whom the zoonosis can cause severe outcomes in immunocompromised individuals.

## Figures and Tables

**Table 1 pathogens-11-01486-t001:** Prevalence of *E. cuniculi* as determined using serological diagnosis methods.

Country		Rabbit Population	Serological Method	*E. cuniculi* Prevalence
Italy	[25]	Pet rabbits	ELISA, CIA	67.2%
	[6]	Farm rabbits	ELISA, CIA	31.6%
	[26]	Pet rabbits	CIA	59.56%
	[17]	Multiple origins (intensive farms, family farms, zoos, research laboratories, pet rabbits)	ELISA	70.5%
UK	[13]	Pet rabbits	ELISA	59%
	[24]	Pet rabbits	ELISA	52%
Austria	[19]	Pet rabbits	IFAT	69.7% (group I) 50% (group II) 100% (group III)
Germany	[27]	Pet rabbits	IFAT, CIA	39.45%
Finland	[28]	Pet rabbits	ELISA	29.2%
Japan	[29]	Pet rabbits	ELISA	63.5% (IgG) 28.5% (IgM)
Taiwan	[20]	Pet rabbits	CIA	63.2%
			ELISA	67.8%
Korea	[30]	Pet rabbits	ELISA	22.6%
Turkey	[10]	Laboratory rabbits	ELISA	49.5%
Brazil	[12]	Pet rabbits	ELISA	81.7%
USA	[21]	Pet rabbits	ELISA	62%

**Table 2 pathogens-11-01486-t002:** *E. cuniculi* prevalences using the diagnosis through molecular genetic techniques.

Country		Rabbit Population	Tissue	PCR Method	*E. cuniculi* Prevalence
Austria	[19]	Pet rabbits	CSF, brain, kidney, heart, liver, lung, spleen, eye globe (only group III)	Conventional PCR	25.4% (groups I and II) 100% (group III)
				Nested PCR	63.6% (group I) 42.1% (group II)
Poland	[32]	Pet rabbits	Urine	RT-PCR	26.21%
Iran	[33]	Pet rabbits	Urine	Semi-nested PCR	32%
Japan	[31]	Pet rabbits	Urine	Nested PCR	7.78% 33.33% 5.6%
		Breeding facilities	Urine		
		Breeding facilities	Feces		
China	[34]	Pet shop rabbits	Feces	Nested PCR	5.8%
Iran	[18]	Pet rabbits	Brain	Nested PCR	59.6% 3.3%
		Laboratory rabbits			
Turkey	[10]	Laboratory rabbits	Eye globe	Conventional PCR	63%

## Data Availability

Not applicable.

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
