# Peer review of "A Review of Encephalitozoon cuniculi in Domestic Rabbits (Oryctolagus cuniculus)—Biology, Clinical Signs, Diagnostic Techniques, Treatment, and Prevention"

_pathogens, 2022, doi:10.3390/pathogens11121486_

Round 1
Reviewer 1 Report
Dear authors
This study aims to review the current data regarding Encephalitozoonosis infection in rabbits, from the pathogenesis, host immunity, clinical signs, and diagnostic methods to treatment and prevention methods of the disease, underlining the significance of limiting the spread of this pathogen among all species.
The manuscript added as a reference for diagnostic of Encephalitozoonosis , also written in proper English. The study is interesting and can be accepted for publication after minor revision for some points presented below:
Comment 1: It is preferable to add the latest statistics on the extent of this disease as well as its impact on the production of rabbits, the subject of the article.
Comment 2: The study did not cover the immunological part sufficiently.
Comment 3: This study should include methods for preventing Encephalitozoonosis (the application of biosecurity).
Comment 4: Line 362: The table title should be moved above the table.
Comment 5: The utilization of the feed additives (such as the influence of dietary supplementation of probiotic bacteria on rabbits challenged with encephalitozoonosis) may decrease or suppress the infection.
Author Response
Dear Editor,
We appreciate all of the constructive criticisms, useful comments and thoughtful suggestions provided by referees. All substantive points and suggestions arising from the referees have been carefully considered during revision of the original manuscript. In all cases, the suggested changes have been carefully considered and in most cases implemented in full.
All revisions made to the original paper have been highlighted in red color for Reviewer #1.
A detailed response to all comments from the reviewers is outlined in the accompanying letter indicated by sentences starting with AU:. The original paper has been extensively revised in accordance with the comments and recommendations arising from the peer-review process, and as a result is much improved.
We hope that the revisions would allow the manuscript to be considered acceptable for publication.
With our best regards,
Anamaria Ioana Paștiu, PhD, DVM
Reviewer #1
This study aims to review the current data regarding Encephalitozoonosis infection in rabbits, from the pathogenesis, host immunity, clinical signs, and diagnostic methods to treatment and prevention methods of the disease, underlining the significance of limiting the spread of this pathogen among all species.
The manuscript added as a reference for diagnostic of Encephalitozoonosis, also written in proper English. The study is interesting and can be accepted for publication after minor revision for some points presented below:
Comment 1: It is preferable to add the latest statistics on the extent of this disease as well as its impact on the production of rabbits, the subject of the article.
AU: We have added in text at Introduction: “Among all parasitic diseases, encephalitozoonosis has proved to be the one most present in rabbits, according to a recent retrospective study on necropsies carried out in Spain [5]. The high morbidity and the potential zoonotic risk of this disease can directly impact the rabbit industry through economical losses, especially in countries where this has become an important meat source, such as Italy or Egypt [6,7]. The high prevalence of the parasite in rabbit farms altogether is most likely due to poor husbandry practices and inadequate health prophylactic measures, while the presence of E. cuniculi in younger animals, between 0 and 4 months old, is probably the result of immature immune systems unable to fight the microorganism [5]. Some recent sources [5,7] support that young rabbits are rather affected than older individuals, respectively females rather than males, but these findings should not be taken as a one-size-fits-all approach.”
Comment 2: The study did not cover the immunological part sufficiently.
AU: We have added in text Host immunity section: “Other essential structures implicated in the cell-mediated immunity of this disease are cytokines, secreted by T-cells as an immune response, which direct macrophages to phagocytose infected cells [4]. IFN-γ is one of these important cytokines that acts as an activator of macrophages, which will lead them to produce toxic oxygen metabolites capable of destroying the phagocytosed E. cuniculi spores. IFN-γ also seems to offer immunologic anti-microsporidial protection regardless of the route of infection [8]. ”
“Intraepitalial lymphocytes (IELs) are another cell type implicated in the cell-mediated immunity, by producing high amounts of IFN-γ and displaying intensive cytolytic processes that can prevent parasite multiplication [8]. Natural killer cells (NK) also have a role by secreting IFN-γ that activate the phagocytic function of macrophages, while also mediating innate responses through perforin-mediated lysis of infected cells [4]. Studies show that dendritic cells may also have an essential protective mechanism, since dendritic cell-deficient mice were susceptible to reinfection with E. cuniculi and sera transfer from young mice exposed to the parasite offered protective immunity [4].”
Comment 3: This study should include methods for preventing Encephalitozoonosis (the application of biosecurity).
AU: We have added in text Treatment and prevention methods section: “Because of the wide spread of the organism and all of its characteristics detailed in this study, this isn’t the simplest thing to accomplish. The first and most essential step would be setting up E. cuniculi-free breeding rabbit colonies, which unfortunately for breeders can require a lot of time and expense. This should be achieved through periodical serological testing of rabbit populations, with the isolation of healthy seronegative individuals from the others in separate cages.”
“It is possible that once the parasite entered a colony, using the “all-in and all-out” system will be necessary, in order to remove the entire population and do a thorough disinfection of the environment before introducing new rabbits. Also, using rabbit breeds from encephalitozoonosis-free rabbitries, such as Lop-eared and Rexes which are thought to be more resistant to the disease, would be ideal to form the new livestock [40].
For rabbits that are not assured to come from E. cuniculi-free colonies, the best way of reducing the spread of infection would be prophylactic administration of fenbendazole orally or in the food, using the same 28-day protocol mentioned earlier. This should also be applied for animals newly acquired before putting them in contact with other rabbits. What needs to be kept in mind is that this method will not guarantee that future infections are impossible to appear [9].”
“Other prophylactic measures that should be applied are reduction of rabbits’ contact with other rabbits or even wildlife,thorough disinfection of the environment using substances like the ones mentioned before, appropriate hygiene through hand-washing and disinfection after handling rabbits and the use of raised food dishes and water bottles that reduce the risk of urine contamination [9, 40].”
Comment 4: Line 362: The table title should be moved above the table.
AU: Done
Comment 5: The utilization of the feed additives (such as the influence of dietary supplementation of probiotic bacteria on rabbits challenged with encephalitozoonosis) may decrease or suppress the infection.
AU: We have added in text Treatment and prevention methods section: “ Supplementation of the diet with probiotic bacteria is also thought to help by increasing the immunity response and by forming a defense against pathogen infection [41], however there are no studies documented on their use against encephalitozoonosis. ”
Reviewer 2 Report
Thank you for opportunity to review the article entitled “A review of the diagnostic techniques of Encephalitozoon cuniculi in domestic rabbits (Oryctolagus cuniculus)”
Overall, this is an interesting review paper about Encephalitozoon cuniculi, an eukaryote, unicellular, spore-forming obligate intracellular microorganism of the phylum Microsporidia. The domestic rabbit is the main host. The paper describes pathophysiology, host immunity, clinical signs, and differentials of the disease in rabbits. Further also diagnostic methods, treatment and prevention options are described.
Minor comments
The title.
Line 2-3: Since the paper describes more than diagnostic techniques the title should be wider e.g “A review of the of Encephalitozoon cuniculi in domestic rabbits (Oryctolagus cuniculus) – a biology, clinical signs, diagnostic techniques, treatment and prevention”
Diagnostic methods.
Line 172-177: Please add reference.
Line 179-183: Please add reference.
Line 276-284. Paragraph not clearly written. The sentence (line 278-280) “They don’t provide information on whether the infection is active, latent or old and cured and they don’t ascertain E. cuniculi as the causative agent of disease.” – oppose/contradict to the sentence (line 281-284) “Most studies report that a high IgM titer indicates early or acute infection, high titers of igG call for a chronic or latent infection and …”. Please rewrite/smooth the paragraph.
Line 365-371: Please add reference.
Line 389: Please add reference to last sentence.
Treatment and prevention methods.
Line 417-422: Please add reference.
Author Response
Pathogens 2058999
A REVIEW OF ENCEPHALITOZOON CUNICULI IN DOMESTIC RABBITS (ORYCTOLAGUS CUNICULUS) – BIOLOGY, CLINICAL SIGNS, DIAGNOSTIC TECHNIQUES, TREATMENT AND PREVENTION
Anca-Alexandra Doboși, Lucia-Victoria Bel, Anamaria Ioana Paştiu*, Dana Liana Pusta
Dear Editor,
We appreciate all of the constructive criticisms, useful comments and thoughtful suggestions provided by referees. All substantive points and suggestions arising from the referees have been carefully considered during revision of the original manuscript. In all cases, the suggested changes have been carefully considered and in most cases implemented in full.
All revisions made to the original paper have been highlighted in green color for Reviewer #2.
A detailed response to all comments from the reviewers is outlined in the accompanying letter indicated by sentences starting with AU:. The original paper has been extensively revised in accordance with the comments and recommendations arising from the peer-review process, and as a result is much improved.
We hope that the revisions would allow the manuscript to be considered acceptable for publication.
With our best regards,
Anamaria Ioana Paștiu, PhD, DVM
Reviewer #2
Thank you for opportunity to review the article entitled “A review of the diagnostic techniques of Encephalitozoon cuniculi in domestic rabbits (Oryctolagus cuniculus)”
Overall, this is an interesting review paper about Encephalitozoon cuniculi, an eukaryote, unicellular, spore-forming obligate intracellular microorganism of the phylum Microsporidia. The domestic rabbit is the main host. The paper describes pathophysiology, host immunity, clinical signs, and differentials of the disease in rabbits. Further also diagnostic methods, treatment and prevention options are described.
Minor comments
The title.
Line 2-3: Since the paper describes more than diagnostic techniques the title should be wider e.g “A review of the of Encephalitozoon cuniculi in domestic rabbits (Oryctolagus cuniculus) – a biology, clinical signs, diagnostic techniques, treatment and prevention”
AU: We have change the title after the reviewer recommendation: A REVIEW OF ENCEPHALITOZOON CUNICULI IN DOMESTIC RABBITS (ORYCTOLAGUS CUNICULUS) – BIOLOGY, CLINICAL SIGNS, DIAGNOSTIC TECHNIQUES, TREATMENT AND PREVENTION
Diagnostic methods.
Line 172-177: Please add reference.
AU: Done
We have added in text (line 186): ”The main methods are the histopathologic diagnosis, the serological diagnosis and the diagnosis through molecular genetic techniques [4].”
Line 179-183: Please add reference.
AU: Done
We have added in text (line 189): “Other paraclinical tests can be helpful in establishing a prognosis, like blood biochemistry to evaluate the renal parameters or computerized tomography (CT) scan to identify the extent of cerebral lesions or differentiate from otitis [15].
Line 276-284. Paragraph not clearly written. The sentence (line 278-280) “They don’t provide information on whether the infection is active, latent or old and cured and they don’t ascertain E. cuniculi as the causative agent of disease.” – oppose/contradict to the sentence (line 281-284) “Most studies report that a high IgM titer indicates early or acute infection, high titers of igG call for a chronic or latent infection and …”. Please rewrite/smooth the paragraph.
AU:We have changed the paragraph: “Although there is a differentiation between the IgM and IgG titers regarding their rise and telling whether the infection is active, latent or old and cured, they don’t ascertain that E. cuniculi is the causative agent of disease [13, 19].”
Line 365-371: Please add reference.
AU: Done
We have added in text (lines383-384): “Some options are hematology for evaluating infection markers, serum protein electro-phoresis for evaluating the specific proteins and biochemistry and kidney ultrasound for assessing renal function [4, 9]. Besides these, a CT scan can be performed to help us evaluate the cerebral damage or lead us to a differential diagnosis of otitis interna/externa [15].”
Line 389: Please add reference to last sentence.
AU: Done
We have added in text (line 403): “This can only be used as an accessory tool in the diagnostic process [36].”
Treatment and prevention methods.
Line 417-422: Please add reference.
AU: Done
We have added in text (lines 432-436): “since this disease has no specific cure and acute cases in immunocompromised animals usually have a deadly outcome [19]. The medication frequently used is symptomatic and antiparasitic drugs, the prognosis being guarded especially in severe brain and kidney lesions that are irreversible [13]. Treating chronic cases of encephalitozoonosis can be easier, but still with no guarantee of a full recovery [4].”